# Restoring Real-World Degraded Events Improves Deblurring Quality

Yeqing Shen
shenyeqing@megvii.com
Megvii
Beijing, China

Shang Li
lishang02@megvii.com
Megvii
Beijing, China

Kun Song
songkun@xs.ustb.edu.cn
University of Science and Technology
Beiing
Beijing, China

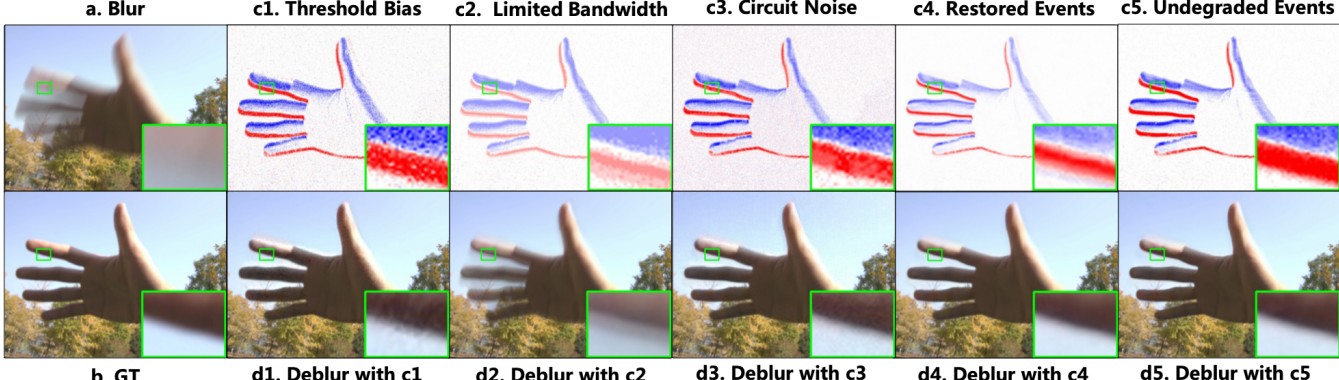

**Figure 1: Different events and the corresponding deblurred results. (a) is the original blurry image, and (b) is the corresponding clear image. (c\*) represents various events, where (c1-3) are three typical degraded events. (c4) is the restoration result of degraded events, and (c5) is undegraded events. (d\*) shows the deblurred results corresponding to the events in (c\*). The deblurring model is DeblurNet trained on undegraded events. The degraded events (c1) with severe threshold bias introduce white and black point artifacts into the deblurred result (d1). The degraded events (c2) with limited bandwidth introduce severe motion blur residue into the deblurred result (d2). The degraded events (c3) with circuit noise introduce significant noise into the deblurred result (d3). Well-restored events (c4) are the output of our event restoration model, contributing to an improvement in the quality of the deblurred results (d4).The deblurred result (d5) corresponding to the undegraded events in (c5) is closest to the ground-truth, reflecting the upper limit of event-based deblurring.**

## Abstract

Due to its high speed and low latency, DVS is frequently employed in motion deblurring. Ideally, high-quality events would adeptly capture intricate motion information. However, real-world events are generally degraded, thereby introducing significant artifacts into the deblurred results. In response to this challenge, we model the degradation of events and propose RDNet to improve the quality of image deblurring. Specifically, we first analyze the mechanisms underlying degradation and simulate paired events based on that. These paired events are then fed into the first stage of the RD-Net for training the restoration model. The events restored in this stage serve as a guide for the second-stage deblurring process. To better assess the deblurring performance of different methods on real-world degraded events, we present a new real-world dataset named DavisMCR. This dataset incorporates events with diverse degradation levels, collected by manipulating environmental brightness and target object contrast. Our experiments are conducted on synthetic datasets (GOPRO), real-world datasets (REBlur), and the proposed dataset (DavisMCR). The results demonstrate that RDNet outperforms classical event denoising methods in event restoration. Furthermore, RDNet exhibits better performance in deblurring tasks compared to state-of-the-art methods. DavisMCR are available at https://github.com/Yeeesir/DVS_RDNet.

*MM '24, October 28-November 1, 2024, Melbourne, VIC, Australia*
© 2024 Copyright held by the owner/author(s). Publication rights licensed to ACM.
ACM ISBN 979-8-4007-0686-8/24/10
https://doi.org/10.1145/3664647.3680714

## CCS Concepts

• **Computing methodologies → Reconstruction**.

## Keywords

Event-based vision, motion deblurring, event degradation, real-world dataset

**ACM Reference Format:**
Yeqing Shen, Shang Li, and Kun Song. 2024. Restoring Real-World Degraded Events Improves Deblurring Quality. In *Proceedings of the 32nd*

*ACM International Conference on Multimedia (MM '24), October 28-November 1, 2024, Melbourne, VIC, Australia.* ACM, New York, NY, USA, 10 pages. https://doi.org/10.1145/3664647.3680714

## 1 Introduction

Dynamic Vision Sensor (DVS) rapidly captures changes in brightness in the environment, and its signals are referred to as events. Due to its high speed and low latency, it is often utilized for motion deblurring. Ideally, high-quality events can accurately record the changes in brightness in motion-blurred regions, guiding event-based methods for better deblurring of motion regions. However, real-world events are degraded, introducing undesirable artifacts into the deblurred results and impacting the final image quality. For example, different pixels in DVS may experience varying degrees of threshold bias[28], resulting in non-smooth black and white points in the deblurred results (Figure 1 (c1) and (d1)). Besides, DVS is constrained by hardware processing speed, leading to event loss and residual motion blur in the deblurred results (Figure 1 (c2) and (d2)). In addition, noise caused by various factors is present in real DVS circuits, and severe circuit noise would introduce undesirable artifacts into the deblurred results (Figure 1 (c3) and (d3)).

The majority of current deblurring methods are exclusively trained on single-degradation events, posing challenges in handling real-world degraded events. They mostly focus on how to handle events for guiding image deblurring[15, 29, 31, 32, 46]. For example, BHA[22] proposes event double integral (EDI), which reconstructs the latent sharp image from the principle of DVS. [46], [15] use CNN to simulate the process of EDI. [29] convert the event into attention map to guide image deblurring. [32] propose an event-enhanced sparse learning network. These studies concentrate on the design of model structures and multi-modal fusion, yet the effectiveness of deblurring is constrained by the degradation inherent in real-world events.

To address this problem, we employ degradation modeling to guide the learning process of event restoration, which is, in turn, employed to improve deblurring quality with the help of RDNet. Specifically, we first model the primary degradation patterns influencing deblurring quality in DVS circuits, categorizing them into threshold bias, limited bandwidth, and circuit noise. Based on this degradation modeling, we construct pairs of undegraded events and degraded events to guide the learning of restoration. Subsequently, we use the proposed RDNet to improve the quality of deblurring. In the first stage, we train an event restoration model with paired event data, employing the rich texture information in blurry images to obtain reliable restored events. In the second stage, we train the deblurring model with paired images. We incorporate the brightness variation with high-temporal resolution from the restored events into the deblurring model, contributing to high-quality deblurred results. In this way, our approach mitigates the potential artifacts introduced by degraded events.

To better assess the deblurring performance of various methods on real-world degraded events, we introduce a real dataset named DavisMCR. Current datasets include only a limited number of event degradation patterns, posing challenges for a comprehensive evaluation of different methods across various degradation scenarios. By contrast, DavisMCR contains events with varying degrees of degradation collected by manipulating the level of environmental

illumination and target objects contrast. Besides, DavisMCR comprises scenes with both simple and complex textures, providing a diverse set of scenarios for evaluating deblurring effects. We employ DavisMCR as a crucial benchmark to assess the performance of various methods in handling diverse degraded real-world events. Moreover, we reconfigure and simulate the event data within the GOPRO dataset. The test set employs simulation settings entirely distinct from the training set, generating a variety of randomly degraded events. We achieve more objective evaluation results on GOPRO by introducing a domain gap in the events data between the training and test sets.

The main contributions of this paper are as follows:

1. We characterize event degradation and create paired data to guide the learning process for event restoration. Subsequently, we propose RDNet, which improves the quality of deblurring by restoring real-world degraded events.
2. We introduce the DavisMCR dataset, which contains events with varying degrees of degradation by manipulating environmental illumination levels and contrasts of target objects. This dataset serves as a comprehensive evaluation platform for assessing the deblurring performance of various methods.
3. RDNet outperform the classic event denoising methods in the event restoration task. Moreover, RDNet exhibits better performance compared to current state-of-the-art methods on synthetic datasets like GOPRO, as well as real-world datasets such as REBlur and DavisMCR.

## 2 Related Work

### 2.1 Event Simulator

The existing DVS datasets are extremely scarce. To enhance the training of deep learning models, event simulators transform large image datasets into annotated event data. ESIM[24] is a classic event simulator widely employed in event-based research[12, 15, 18, 28, 29, 31, 36, 39, 46] for data preparation. ESIM proposes a paradigm of rendering continuous frames using a single image and camera trajectory. It simulates the events and employs an adaptive sampling strategy to mimic the signal characteristics of DVS under varying brightness conditions. However, ESIM only accounts for simple noise simulation caused by thresholding and does not simulate the complex degradation in actual DVS circuits. As a result, there exists a certain gap between the simulated events and real-world degraded events. Vid2E[7] explores variations in time sampling rates and circuit parameter augmentations based on ESIM. V2e[10] further simulates various characteristics of DVS circuits, including pixel-level Gaussian event threshold mismatch, finite intensity-dependent bandwidth, and intensity-dependent noise. With novel insights into sensor design and physics, DVS-Voltmeter[14] considers voltage variations in DVS circuits to account for the randomness caused by photon reception, as well as noise effects from temperature and parasitic photocurrent. It models the event generation process as a more comprehensive stochastic process. The aforementioned methods provide effective tools for the current research, in which we model event degradation and construct paired events on the foundation of the event simulator. These paired events serve as a guide for the learning of the event restoration.

## 2.2 Motion Deblurring

**Image-only Deblurring.** The mainstream image-only deblurring methods learn blurry features through deep learning models and then output clear images [2, 3, 5, 6, 9, 13, 20, 23, 25, 26, 30, 40–43, 48]. For example, [5, 30, 42] employ the coarse-to-fine approaches, gradually restoring clear images at different resolutions in the pyramid. [2, 3] design backbone models from the perspective of network architecture. DeblurGAN[13], based on conditional GAN, achieves optimization in both structural similarity metrics and visual appearance. SPAIR[23] utilizes the non-uniformity of severe degradation in the spatial domain and proposes a learning-based universal solution for recovering images subjected to spatially varying degradation. MPRNet[41], through a multi-stage architecture, progressively learns the restoration function of the degraded input, achieving a balance between spatial details and high-level contextual information. Restormer[40] incorporates critical design in building blocks (multi-head attention and feedforward networks) to capture long-range pixel interactions, making it suitable for large images by applying a transformer. MSGD[25] introduces an image-conditioned Dynamic Programming Method, employing multiscale structure guidance as an implicit bias to guide image deblurring. MRLPFNet[6] proposes a learnable low-pass filter utilizing a self-attention mechanism to model low-frequency information, alongside an additional fully convolutional neural network employing standard residual learning to model high-frequency information. However, these image-only deblurring methods merely learn blur patterns from the images that lack motion information, and this results in insufficient generalizability of these methods. In scenarios with severe motion blur, image-only deblurring methods typically exhibit weaker performance compared to event-based deblurring.

**Event-based Deblurring.** The events containing changes in brightness with high-temporal resolution provide guidance for deblurring images in dynamic scenes. Event data is a distinct modality from images and it features spatiotemporal sparsity. Current event-based deblurring methods focus on integrating these two modalities to achieve image deblurring [1, 4, 8, 11, 12, 19, 21, 22, 27, 29, 31, 32, 34, 35, 37, 38, 44–47]. BHA[22] starts from the working principles of DVS, employing double integration on events to reconstruct the brightness of each pixel in the image. It elucidates the principles of event-based deblurring and demonstrates feasibility as shown in the experimental results. However, this approach introduces noise from events into the deblurred results, leading to severe artifacts. Following this line of reasoning, further studies attempt to improve the deblurring performance by optimizing the models. EVDI[46] simulates the double integral calculation through a convolutional neural network and uses the network to simulate the process of BHA. LEMD[11] introduces a sequential formulation for event-based motion deblurring and elucidates its optimization with a deep network. D2Nets[27] utilizes a bidirectional LSTM detector to identify the nearest sharp frame , upon which deblurring is subsequently performed. EBKE[19] proposes a novel approach for estimating blur kernels based on events, enabling the recovery of complex blur motions through the utilization of blur kernels. Because this method is not relying on machine learning methodologies, it operates independently of training data. HDN[44] adopts a recurrent encoder-decoder architecture to generate dense cyclic event representations,

encoding the entirety of historical information, followed by deblurring processing. EvIntSR-Net[8] aligns the domains between event streams and intensity frames, learning to fuse latent frame sequences in a recurrently updated manner. DS-Deblur[38] proposes a dual-stream based event-image fusion framework for motion deblurring, adaptively aggregates the frame and event progressively at multiple levels. EIFNet[37] proposes an event-image fusion network that is grounded on modality-aware decomposition and recomposition techniques, facilitating enhanced integration of features from both event and image modalities. ERDNet[1] learns event-based motion deblurring with a residual learning approach. NEID[4] proposes a non-coaxial event-guided deblurring method that spatially aligns events to images while refining the image features from temporally dense event features. RED-Net[35] employs simultaneous estimation of optical flow and latent images, leveraging blur consistency and photometric consistency to enable self-supervision of the deblurring network using real-world data. SAN[47] proposes a scale-aware network capable of adapting to various spatial and temporal resolutions of motion blur. ESL-Net[32] uses the framework of sparse learning to restore low-quality images. [31] added modulated deformable convolutions to the model to make better use of dense motion information. UEVD[12]introduces a novel exposure time-based event selection method. It selectively utilizes event features by estimating cross-modal correlations between blurry frame characteristics and events. EFNet[29] converts events into attention maps to guide image deblurring. These studies provide valuable insights into model design for event-based deblurring. However, they lack the ability to handle the degradation in real-world events, resulting in a lack of robustness in practical applications. Therefore, we propose RDNet, which improves deblurring quality through event restoration.

## 3 Methodology

### 3.1 Problem Definition

The principle of DVS can be expressed by Eq.1, where $I_t$ represents the image at $t$ time, $p \in \{-1, +1\}$ indicates the polarity of the event signal, and $c$ indicates the threshold of DVS.

$$log(I_{t+\Delta t}) - log(I_t) = p \cdot c \tag{1}$$

We define the event signal as $e_{xy}(t) = p \cdot \delta(t - t_0)$, which means that an event signal with polarity $p$ is generated at $(x, y)$ at the time of $t_0$, and $\delta(\cdot)$ is an impulse function. Blurry image $B(t_f)$ is the integral of image $I(t)$. The final deblurred result can be calculated by Eq.(2). The detailed derivation of Eq.2 can be found in the supplementary material.

$$I[t_r] = exp(log(B[t_f]) - log(\sum_{n=0}^{N} exp(c \sum_{i=0}^{n} e[i]))) \tag{2}$$

In order to facilitate learning event-based deblurring with a CNN-based network, we transform the original event data and pass the original event quadruple $(t, x, y, p)$ through Eq.3 into a tensor of $H \times W \times N_e$. In Eq.3, $t_n = t_0 + n \cdot T/N_e$, $T$ is exposure time.

$$E(h, w, n) = \int_{t_n}^{t_{n+1}} e_{wh}(s)ds \tag{3}$$

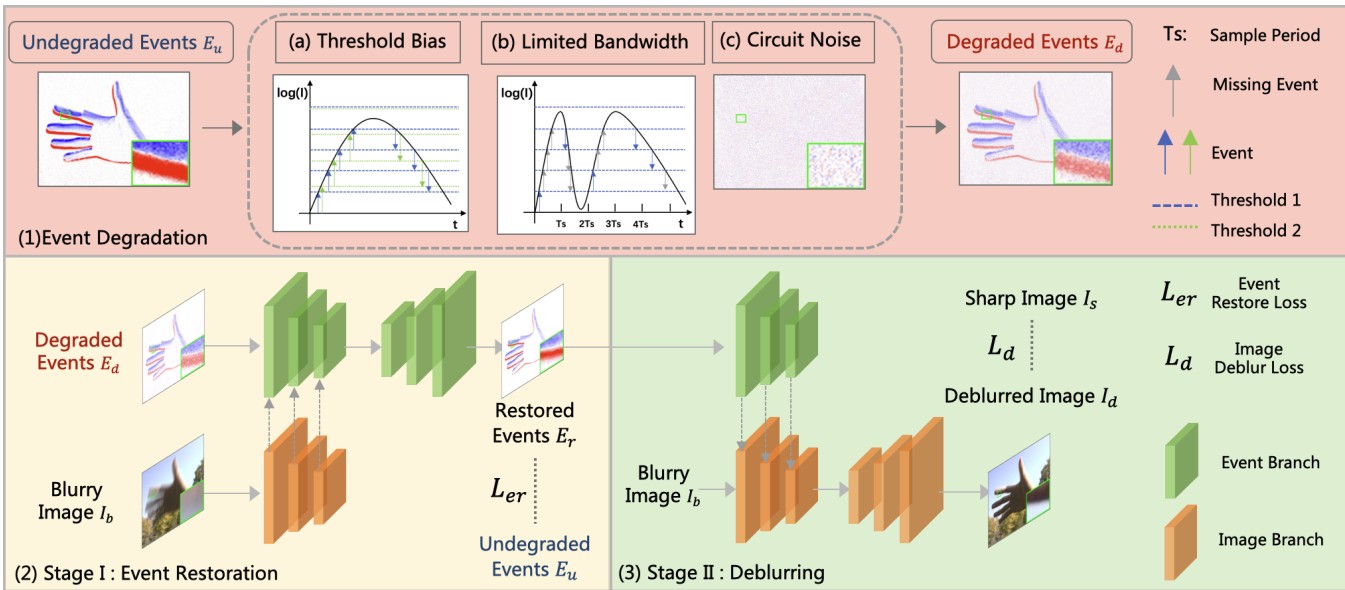

**Figure 2: The event degradation process and the pipeline of RDNet. The red region (1) above illustrates the event degradation process for constructing paired data of undegraded $E_u$ and degraded events $E_d$. (a) illustrates how threshold bias introduces differences in events. (b) represents how limited bandwidth leads to event loss. (c) provides visualization of simulated circuit noise. The yellow region (2) below is the first-stage event restoration. Degraded events $E_d$ and blurry image $I_b$ are fed into dual-branch encoders, and a single-branch event decoder generates the restored event $E_r$. The ground-truth is undegraded event $E_u$, and the loss is $L_{er}$. The green region (3) below is the second-stage event-based deblurring. Restored event $E_r$ and blurry image $I_b$ are fed into dual-branch encoders, and a single-branch image decoder generates the deblurred image $I_d$. The ground-truth is sharp images $I_s$, and the loss is $L_d$.**

Blurry image and sharp image are defined as 3D tensors $I_b$ and $I_s$, with dimensions $H \times W \times 3$.

## 3.2 Event Degradation Modeling

Based on the impact of degradation in real-world events on image deblurring, we categorize degradation patterns into threshold bias, limited bandwidth, and circuit noise. By employing degradation modeling, paired data consisting of degraded and undegraded events can be constructed to guide the learning process of event restoration.

**Threshold Bias.** When there is a bias in the threshold, the number of triggered events in DVS deviates accordingly, and this leads to significant fluctuation in the event waveforms in motion regions (Figure 1 (c1)). However, the deblurring model still performs deblurring based on the default standard thresholds during training, leading to outliers in the results, as depicted by the non-smooth black and white points in Figure 1 (d1). From Eq.1, it can be derived that the threshold determines the change ratio in brightness that triggers event signals for each pixel. As shown in Figure 2 (a), the black curve represents the log-domain brightness signal, and the blue and green dashed lines represent threshold1 and threshold2, respectively. These two thresholds with tiny difference triggers different numbers of event signals for the same black brightness curve (Green: 3 positive events, 2 negative events; Blue: 4 positive events, 2 negative events). This illustrates that small threshold bias results in significant discrepancies in events.

**Limited Bandwidth.** While DVS is renowned for its high-speed recording of environmental changes, real devices are constrained by hardware processing speed, resulting in limited bandwidth. As depicted in Figure 1 (c2), when DVS bandwidth is limited, fewer events are triggered compared to undegraded conditions. The deblurring model is unable to accurately restore the changes in brightness in motion regions, resulting in residual motion blur in the Figure 1 (d2). When the ambient changes in brightness rapidly, the finite bandwidth leads to the loss of events. As illustrated in Figure 2 (b), the black curve represents the log-domain brightness signal, and the blue dashed line represents the threshold. $T_s$ represents the sampling period, which reflects the bandwidth. One event is generated within one sampling period at most. Consequently, only the blue arrows are sampled, while the gray arrows are not. Despite the fact that the actual brightness variation triggers 7 positive events and 6 negative events, only 2 positive events and 3 negative events are captured due to bandwidth limitations.

**Circuit Noise.** The deblurring model demonstrates some denoising capability for spatially sparse background noise. However, in the presence of severe noise, distinguishing between noise signals and valid signals becomes a challenging task. As illustrated in Figure 1 (d3), severe circuit noise introduces undesirable artifacts into the deblurred results. In the photodetection process, the quantum properties of photons result in granular noise[10], which is particularly severe under low-light conditions and is commonly modeled as

a Poisson process. Leakage noise events caused by junction leakage and parasitic photocurrent during the reset switch transition of the DVS pixel lead to the noise. DVS sensors always exhibit some hot pixels that continuously trigger events at a high rate even in the absence of input. These hot pixels may arise from reset switches with exceptionally low thresholds or with very high dark currents[10]. Therefore, we introduce leakage noise and hot pixel noise into the noise modeling process. While there are other noise-contributing factors in real DVS circuits with effects similar to those on deblurring results, we do not offer further analysis here. Randomly generated event noise is shown in Figure 2 (c). It can be observed that in motionless scenes, a substantial number of events are generated.

## 3.3 Two-Stage Pipeline

The framework of RDNet is illustrated in Figure 2 (2, 3). We propose a two-stage pipeline to improve the quality of deblurring in the second stage through the restoration of degraded events in the first stage.

In the first stage, we employ a model structure consisting of a dual-branch encoder and a single-branch event decoder. Initially, the degraded events $E_d$ are fed into the event encoder, while the blurry image $I_b$ is fed into the image encoder. Subsequently, the features generated by the image encoder are added to the features generated by the event encoder at the same scale, incorporating rich texture and color information from the image into the event branch to guide accurate restoration of events. Finally, the latent features generated by the encoder are fed into the event decoder to obtain the restored events $E_r$. Model training is supervised with undegraded events $E_u$, and the first term of the event restoration loss $L_{er}$ is employed.

In the second stage, we also employ a model structure with a dual-branch encoder and a single-branch image decoder. Initially, the restored events $E_r$ are fed into the event encoder, and the blurry image $I_b$ is fed into the image encoder. Subsequently, the features generated by the event encoder are added to the features generated by the image encoder at the same scale, integrating brightness variation with high-temporal resolution from the event branch into the image branch to guide deblurring. Finally, the latent features generated by the dual-branch encoder are fed into the image decoder to obtain the deblurred image. Model training is supervised with the sharp image, and the loss function employed is $L_d$. Additionally, the second term of the loss in the second stage is derived from $L_{er}$. Details on the structure of RDNet can be found in the supplementary.

**Loss Function.** The loss $L_{er}$ employed to supervise event restoration is computed according to Eq.4, where $\alpha$ and $\beta$ are hyperparameters. The first term is the L1 norm between the restored events $E_r$ and undegraded events $E_u$, directly utilized to supervise the training in the first stage. The second term is the L1 norm of the latent features of $E_r$ and $E_u$, where $F(\cdot)$ represents the event encoder used in the second stage. This term is employed during the second stage of training to finetune the restoration model. As events are sparsely distributed in space and time, $L_{er}$ is computed only at the event response locations, which are the positions where the value of undegraded events $E_u$ and degraded events $E_d$ are non-zero.

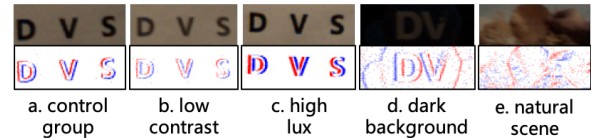

a. control group  b. low contrast  c. high lux  d. dark background  e. natural scene

**Figure 3: The innovation of DavisMCR dataset. (a) represents the control group, capturing a normal contrast text motion scene under the illumination of lux=800. The events exhibit clear textures with minimal noise. (b) depicts a low-contrast text motion scene, where events are relatively weak, and the edges are less defined. (c) showcases a text motion scene captured in a high-lux environment, displaying events with clear edges and minimal noise. (d) presents a text motion scene with a dark background, showing events with severe background noise. (e) illustrates a natural scene with events containing diverse forms and various intensity levels.**

$$L_{er} = \alpha \cdot L1((E_r - E_u)) + \beta \cdot L1(F(E_r) - F(E_u)) \qquad (4)$$

The loss for supervising event-based deblurring, denoted as $L_d$, is calculated with the L1 norm between the deblurring image $I_b$ and the sharp image, as described in Eq.5. This loss is applied during the training of the second stage.

$$L_d = L1((I_d - I_s)) \qquad (5)$$

## 4 DavisMCR Dataset

We propose a real-world dataset DavisMCR, comprising event data captured in varying environmental brightness and target object contrast conditions. Data collection is performed using the DAVIS 346c camera, and the ColorSpace CS-HDR-MFS lightbox is employed to set the ambient brightness.

Figure 3 (a) represents the control group, a text scene with normal contrast captured at a brightness of lux=800. The event texture is clear, and the noise is low. To generate scenes with different contrast ratios in a real environment, we print versions of the same scene with varying transparencies. In scenes with lower contrast, the triggered event signals are more sparse, while scenes with higher contrast exhibit denser event signals. As shown in Figure3 (b), the events in the low-contrast text motion scene are weak, and the edges are not clear. To generate scenes with different brightness levels in a real environment, we capture targets with brightness ranging from lux=100 to lux=10000. Across different brightness levels, we keep the aperture size constant and adjust the exposure time of the APS image to avoid overexposure. Due to the asynchronous readout of DVS and APS circuits in DAVIS, the exposure time of APS does not affect the DVS signal. This implies that, under a constant aperture, a brighter environment allows more light to enter per unit time. In other words, in brighter scenes, the signal-to-noise ratio of the induced current in the DVS circuit is higher. As seen in Figure 3 (c), the text motion scene captured in a high-lux environment features clear edges with low noise. To obtain events with different noise levels, we select scenes with different background brightness. Events of regions with bright backgrounds have less noise, and events of regions with dark backgrounds have more noise. As shown in Figure 3 (d), events captured in the dark-background text motion scene exhibit severe background noise. To

comprehensively evaluate the deblurring results, we select multiple scenes as targets for capturing. The deblurred details on simple textures are easy to observe and discern as shown in Figure 3 (a). By contrast, complex textures closely resemble real-world scenarios as depicted in Figure 3 (e). More details about the DavisMCR dataset are shown in the supplementary material.

## 5 Experiment

### 5.1 Dataset

Our training set is the synthetic dataset GOPRO[17], using v2e[10] for event simulation. Our test set includes the synthetic dataset GOPRO[17], the real dataset REBlur[29], and DavisMCR. The test set of GOPRO employs simulation settings entirely distinct from the training set, encompassing a variety of randomly degraded events. We, therefore, achieve more objective evaluation results on GOPRO by introducing a domain gap in the events data between the training and test sets. More details about the dataset can be seen in the supplementary materials.

### 5.2 Implementation details

We crop the images of GOPRO data into 256×256 size, and the corresponding events are also cropped into the same size after being divided into channels. We set the parameters $N_e$ as 10, indicating that events within the exposure time are divided into 10 channels. The rationale is that an inadequately small $N_e$ can lead to the loss of temporal information, while an excessively large $N_e$ would result in sparse events, increasing the difficulty of network training. Both $\alpha$ and $\beta$ in $L_{er}$ are set to 0.5. We use Adam with an initial learning rate of $2 \times 10^{-4}$ and a cosine learning rate strategy with a minimum

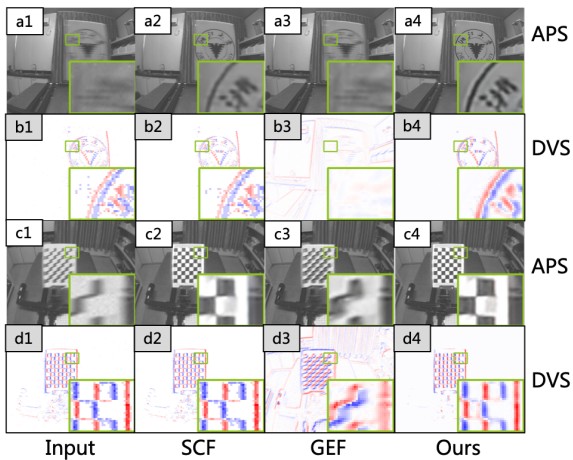

**Figure 4: Comparision of restoring results on REBlur. The first column consists of input blurry images and their corresponding degraded events. The second column shows restored events obtained by SCF and the corresponding deblurred results. The third column presents restored events obtained by GEF and the corresponding deblurred results. The fourth column displays restored events obtained by the first-stage of RDNet and the corresponding deblurred results.**

learning rate of $10^{-7}$. The model is trained for 300k iterations with a batch size of 8. We use the same experimental setup for training in both stages. To facilitate subsequent comparisons, we introduce DeblurNet as a baseline model, which shares the same structure as the second-stage RDNet. To ensure a fair comparison, DeblurNet is designed with an equivalent number of parameters to the two-stage RDNet.

### 5.3 Results of Event Restoring

As events are sparse and discontinuous both in space and time, and there is a lack of accurate ground-truth for events, our comparisons are conducted qualitatively on real datasets REBlur[29] and DavisMCR. We compare the event restoration results of the first stage of RDNet with two classic event denoising methods, SCF[16] and GEF[33]. As shown in Figure 4, (a1) and (c1) are the input images, while (b1) and (d1) are the input original events. (b2-b4) and (d2-d4) show the event results restored by different methods, and (a2-a4) and (c2-c4) show the deblurred results using the corresponding restored events. To facilitate a more effective comparison of event restoration results, the restored events of SCF and GEF are fed into a DeblurNet to obtain deblurred results. This DeblurNet is trained with undegraded events.

As shown in Figure 4, comparing the input events (b1, d1) with the output of SCF (b2, d2), it can be observed that SCF can only denoise relatively discrete event noise in space, exhibiting weak event restoration capability. This leads to some residual motion blur in the deblurred results (a2, c2). Comparing the input events (b1, d1) with the output of GEF (b3, d3), it can be seen that the reconstruction of GEF relies on the image texture. When motion blur is severe, the restored events lose texture, leading to lots of residual motion blur in results (a3, c3). Additionally, GEF may introduce artifacts by erroneously restoring events in non-motion regions with strong textures. Comparing the input events (b1, d1) with the output of RDNet (b4, d4), it is evident that RDNet can effectively restore events at the motion regions and recover events in a smooth, high-quality fashion. RDNet, based on the high-quality events restored in the first stage, achieves deblurred results with fewer residual motion blur and artifacts.

Experimental results on the DavisMCR dataset can be found in the supplementary material.

### 5.4 Results of Deblurring

We quantitatively and qualitatively evaluate our RDNet and the other deblurring methods on GOPRO, REBlur, and DavisMCR. We compare our method with state-of-the-art image-only and event-based deblurring methods. For image-only methods[2, 3, 5, 13, 23, 30, 40–42], we use the official release model for results testing. For MSGD[25] and MRLPFNet[6], since no reproducible models or parameters are public, only the numerical results on the GOPRO dataset are compared. For event-based methods[29, 32], we retrain according to the official settings in order to offer a fair comparison.

*5.4.1 Evaluation on GOPRO Dataset.* We report deblurred results on GOPRO dataset in Table 1. Compared to the best existing image-only methods MRLPFNet[6] and event-based method EFNet[29], our method achieves 3.32 dB and 1.72 dB improvement in PSNR and 0.012 and 0.007 improvement in SSIM, respectively. These

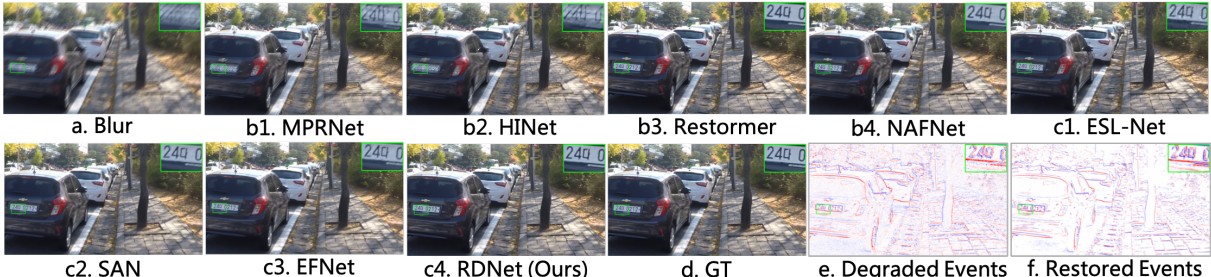

**Figure 5: Results of deblurring on GOPRO dataset. (a) is the input blurry image. (b*) are the results of image-only deblurring methods. (c*) are the results of event-based deblurring methods. (d) is the ground-truth. (e) are input degraded events. (f) are restored events.**

image-only methods lack high temporal resolution information from events, making it challenging for them to accurately estimate the blur process and perform image deblurring. Existing event-based methods, due to their lack of capability to recover degraded events, introduce artifacts in the deblurring results, leading to lower numerical outcomes. The two-stage RDNet effectively restores degraded events and combines high temporal resolution information from events to achieve accurate deblurring.

We show qualitative results of deblurring on GOPRO in Figure 5. As shown in Figure 5 (b1, b2), image-only methods MPRNet[41], HINet[3] cannot accurately deblur the blurry image. Instead, they generate a lot of ghosts in deblurred results. Restormer[40] and NAFNet[2] achieve better visual effects as shown in Figure 5 (b3,

**Table 1: Deblurring results on GOPRO. Note: As the results of image-only methods are unaffected by events, all results of these methods are directly adopted from previous studies. To ensure fairness in comparison, the results of all event-based methods are based on retrained models.**

| Method | Params(M) | PSNR | SSIM |
|---|---|---|---|
| *Image-only* | | | |
| DeblurGAN[13] | 4.30 | 28.70 | 0.858 |
| SRN[30] | 10.25 | 30.26 | 0.934 |
| DMPHN[42] | 7.23 | 31.20 | 0.940 |
| SPAIR[23] | - | 32.06 | 0.953 |
| MIMO-UNet[5] | 6.81 | 31.73 | 0.951 |
| MPRNet[41] | 20.13 | 32.66 | 0.959 |
| HINet[3] | 88.67 | 32.77 | 0.959 |
| Restormer[40] | 26.13 | 32.92 | 0.961 |
| MSGD[25] | 30.0 | 33.20 | 0.963 |
| NAFNet[2] | 67.86 | 33.71 | 0.966 |
| MRLPFNet[6] | 20.6 | 34.01 | 0.968 |
| *Event-based* | | | |
| BHA[22] | - | 28.23 | 0.921 |
| eSL-Net[32] | 0.18 | 33.52 | 0.954 |
| SAN[47] | 2.25 | 35.53 | 0.968 |
| EFNet[29] | 8.47 | 35.61 | 0.973 |
| **RDNet (Ours)** | 7.86 | **37.33** | **0.980** |

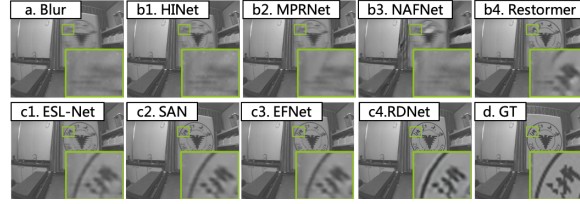

**Figure 6: Comparision of deblurred results on REBlur. $b*$ are the results of the image-only methods. $c*$ are the results of event-based methods. $d$ is the ground-truth. The results of RDNet in $c3$ have clearer textures and fewer artifacts.**

b4). Both methods restore clear edges and generate no additional artifacts. As shown in Figure 5 (e), the events in the flat area contain a lot of noise, and the events on the edge are not completely continuous due to the threshold bias. Event-based methods eSL-Net[32], SAN[47] and EFNet[29] are also affected by degraded events as shown in Figure 5 (c1, c2, c3). There are some abnormal color pixels in the deblurred images, which make the edge blurry and the flat areas noisy. By contrast, as shown in Figure 5 (f), the event noise in the flat area is effectively suppressed, and the edges are clear and continuous. As a result, the output of RDNet in Figure 5 (c4) has sharp edges and few artifacts.

*5.4.2 Evaluation on REBlur Dataset.* We report deblurred results on REBlur dataset in Table 2. Compared to the best existing image-only method Restormer[40] and event-based method EFNet[29], our method achieves 1.92 dB and 0.83 dB improvement in PSNR and 0.021 and 0.006 improvement in SSIM, respectively. Notably, NAFNet[2] exhibits a significant degradation in performance, reflecting its poor generalization to data in different domains.

We show qualitative results on REBlur in Figure 6. As shown in Figure 6 (b1, b2, b3), HINet, MPRNet, and NAFNet show no significant deblurring effects when directly tested on the REBlur dataset. As depicted in Figure 6 (b4), Restormer is capable of partially restoring the shape of texture. However, considerable motion blur residues persist, and some details of the texture remain unclear. In Figure 6 (c1, c2, c2), event-based methods, ESL-Net, SAN and EFNet, exhibit evident deblurring effects in motion regions. These methods stably recover text, with only partial motion blur residues in dense texture areas. As shown in Figure 6 (c4), RDNet achieves a

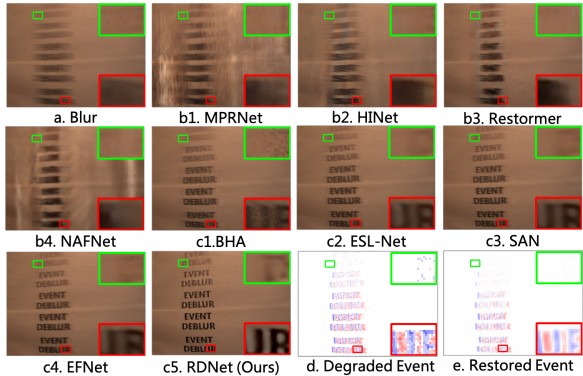

Figure 7: Comparision of deblurred results on DavisMCR. $a$ is the input blurry image. $b*$ are the results of the image-only methods. $c*$ are the results of event-based methods. $d$ is the degraded event of the input and $e$ is the restored event. Compared with the results of other methods, the results of RDNet in $c4$ have clearer textures and fewer artifacts.

clearer restoration of texture, which is closest to the texture of the ground-truth. More experimental results on the REBlur dataset can be found in the supplementary material.

*5.4.3 Evaluation on DavisMCR Dataset.* We show qualitative results on DavisMCR in Figure 7. As shown in the Figure 7 (b*), image-only methods exhibit poor performance. When the domain gap between the real data and the training data is large, the deblurring performance deteriorates seriously. As shown in the Figure 7 (d), text with different contrast produces events with different degradation conditions. The events on the top row are so sparse that the texture cannot be seen clearly. Besides, there are many discontinuous events at the edge of the motion region due to the threshold bias. As shown in the Figure 7 (c1-4), these sparse events lead to weak deblurring effects in the existing methods, and discontinuous events in the motion region cause unclear restored edges.

Table 2: Deblurring results on REBlur. Note: To ensure a fair comparison of the generalization performance of different methods, we tested models trained on the GOPRO dataset.

| Method | Params(M) | PSNR | SSIM |
|---|---|---|---|
| *Image-only* | | | |
| MIMO-UNet[5] | 6.81 | 29.14 | 0.900 |
| MPRNet[41] | 20.13 | 33.86 | 0.946 |
| HINet[3] | 88.67 | 33.76 | 0.942 |
| Restormer[40] | 26.13 | 34.39 | 0.953 |
| NAFNet[2] | 67.86 | 27.80 | 0.822 |
| *Event-based* | | | |
| eSL-Net[32] | 0.18 | 35.20 | 0.963 |
| SAN[47] | 2.25 | 35.57 | 0.968 |
| EFNet[29] | 8.47 | 35.48 | 0.968 |
| **RDNet (Ours)** | 7.86 | **36.31** | **0.974** |

Table 3: Ablation Study on different components of RDNet.

| | Training Events | Model | Testing Events | PSNR | SSIM |
|---|---|---|---|---|---|
| 1 | Undegraded | DeblurNet | Degraded | 35.74 | 0.967 |
| 2 | Degraded | DeblurNet | Degraded | 36.81 | 0.971 |
| **3** | **Both** | **RDNet** | **Degraded** | **37.33** | **0.980** |
| 4 | Undegraded | DeblurNet | Undegraded | 37.96 | 0.985 |

By contrast, the result of RDNet has many details and clear edges as shown in the Figure 7 (c5). It is worth noting that the events in the green box above Figure 7 (e) are discarded, but those in Figure 7 (c5) still exhibit evident deblurring effects. This indicates that in areas with particularly sparse events, RDNet can achieve effective deblurring by relying solely on the information present in the image.

## 5.5 Ablation Study

In order to verify the effectiveness of the degraded event data and the proposed RDNet, we conduct comparative experiments on the GOPRO dataset. In the experiment, we adopt different data and different deblurring models. As shown in Table 3-setting1, we use undegraded events to train DeblurNet and test with degraded events. Compared with setting-1, setting-2 replaces the training events with degraded events and obtains a PSNR improvement of 1.07dB, which illustrates the effectiveness of event degradation modeling. Setting-3 is the experimental setting of RDNet. Compared with setting-2, there is an improvement of 0.51 dB in PSNR. The RDNet is trained with both degraded and undegraded events in the first stage. During this supervised training, undegraded events provide additional information for RDNet, guiding the model to restore degraded events, thereby improving the quality of deblurring. Setting-4 is a set of ideal experiments. Both the training and test set are undegraded events. It finally reaches 37.96 dB in PSNR, which demonstrates the upper limit that event-based deblurring methods can achieve when events are not degraded.

The above experiments demonstrate the effectiveness of the degraded event data and the RDNet. We also test the above model on the real dataset REBlur and DavisMCR, and the qualitative results are in the supplementary material.

## 6 Conclusion

To reduce the artifact caused by real-world event degradation on deblurring results, we first characterize event degradation and create paired data to guide the learning process for event restoration. Subsequently, we use RDNet to improve the quality of deblurring by restoring degraded events. In addition, we propose a real dataset named DavisMCR, comprising event data captured under varying environmental brightness and target object contrast conditions. This dataset leverages real-world degraded events, which serves as a platform for assessing the deblurring performance of various methods. The results demonstrate that RDNet attains high-quality deblurring performance on both synthetic and real datasets through event restoration. In the future, we plan to extend RD-Net to tasks such as event-based video frame interpolation and super-resolution.

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
