# OpenReview forum: "Restoring Real-World Degraded Events Improves Deblurring Quality"
_acmmm.org/ACMMM/2024/Conference — MM2024 Poster_

### Official Review · Reviewer_MQzT · 2024-05-17

**Rating:** 4
**Confidence:** 2

**Summary:**

Existing event-based image deblurring approaches focus on designing the network structures and multi-modality fusion units, neglecting the intrinsic degradation in real-world events. This paper designs a two-stage framework, where the first stage aims to restore the degraded events and the second stage incorporate the prior from the event restoration model into the image deblurring model.

**Strengths:**

1.  The paper takes the degradations in the real-world events into consideration, thus attempting to firstly restoring the degraded events and then apply the event restoration model to guide the image deblurring.

2. This paper develops a DavisMCR dataset, including extensive events with diverse degradation levels by manipulating environmental illumination and contrasts.

**Limitations:**

1. The reviewer are confused whether the event branch in the second stage is fixed. If the image branch in the second stage is the pre-trained model from the first stage and whether it is updated in the second stage. The authors are encouraged to provide these details in Figure 2 and manuscript for clarity.

2. This paper decomposes the event restoration and image deblurring into two stages. Why not formulate them into an end-to-end framework in a mutual learning manner? The reviewer expects the authors to discuss the motivation and purpose behind adopting a two-stage pipeline.

**Suitability:**

2

---

### Official Review · Reviewer_mVsK · 2024-05-24

**Rating:** 5
**Confidence:** 3

**Summary:**

This paper proposes a method, named RDNet, to improve the quality of image deblurring by restoring real-world degraded events captured by Dynamic Vision Sensors (DVS). The authors first analyze and model the degradation patterns in DVS events, including threshold bias, limited bandwidth, and circuit noise. They then construct paired degraded and undegraded events to train the first stage of RDNet for event restoration. The restored events are subsequently used to guide the second stage of RDNet for image deblurring. Additionally, the authors introduce a new real-world dataset, DavisMCR, containing events with varying degrees of degradation for comprehensive evaluation.

**Strengths:**

1.The paper proposes a novel approach to improve image deblurring quality by addressing the degradation issue in real-world DVS events, which has been largely overlooked in previous works. The idea of restoring degraded events before deblurring is creative and well-motivated.
2.The authors provide a thorough analysis of the degradation mechanisms in DVS events and model them accordingly. The two-stage RDNet architecture, with the first stage dedicated to event restoration and the second stage for deblurring, is technically sound and well-designed.
3.The introduction of the DavisMCR dataset, containing real-world events with varying degradation levels, is a valuable contribution to the community. It provides a comprehensive testbed for evaluating deblurring performance under different degradation scenarios.

**Limitations:**

1.Although the DavisMCR dataset is a valuable contribution, it would be beneficial to provide more details on the diversity of scenes and motion patterns it covers, as well as potential limitations or biases in the dataset.
2.The degraded and undegraded event pairs used for training the event restoration module were generated using v2e. A section comparing the degradation characteristics of real events and the simulated degradation from v2e should be added, analyzing the differences between them. While the simulated results from v2e are quite realistic, there may still be domain gaps between simulated and real events, and some analysis should be provided to avoid potential misunderstandings for the readers.
3.The degradation of events shown in the provided images is not apparent. Referring to the paper "arXiv2024 Seeing Motion at Nighttime with an Event Camera," events exhibit noticeable trailing artifacts in low-light conditions. Does the proposed method effectively correct the delayed or trailing events from event cameras in low-light scenarios?
4.More results on real-world events could be provided, along with no-reference perceptual quality metrics to demonstrate the superiority of the proposed method. Showing too many images and tables from the GOPRO dataset and REBlur is not very informative, as these datasets do not contain real events and the events are simulated from sharp frames. This provides additional information, but comparing against methods like Restormer is unfair.

**Suitability:**

3

---

### Official Review · Reviewer_X7qP · 2024-05-25

**Rating:** 2
**Confidence:** 3

**Summary:**

The paper addresses image deblurring from an event-based method，by modelling the degradation of events and propose RDNet to improve the quality of image deblurring. Furthermore，this paper presents a real-world dataset named DavisMCR.

**Strengths:**

1. In this paper, DavisMCR dataset is proposed for evaluating the performance of different methods in dealing with real degradation events.

2. The experiments in the paper show that RDNet achieves good performance on both event recovery and deblurring tasks.

**Limitations:**

1. The proposed method is too simple and the innovation is limited. Specifically, the conventional two-stage scheme and the encoder-decoder structure is adopted, and the innovation is only the introducing of events information.

2. More new events-based methods need to be added for comparison in the experimental section

3. The authors compared two event recovery methods, SCF and GEF, in Figure 4. Can further compare the differences with undegraded events in the REblur dataset? Additionally, could the deblurring results be compared with the visual quality of the ground truth images? Because visually it seems that the deblurring results obtained from events recovered using the SCF method also look good.

4. To facilitate a more effective comparison of event restoration results, the restored events of SCF and GEF are fed into a DeblurNet to obtain deblurred results. This DeblurNet is trained with undegraded events. However, in the ablation experiments it was illustrated that the RDNet was trained using both degraded and undegraded events. It was indicated that the results were better than using only degraded events or only undegraded events. Therefore, it would be advisable to feed the recovered events from SCF and GEF into the second stage of RDNet in order to compare the results of the deblurring, thus confirming that RDNet has a stronger event recovery capability. It is hoped that the author can provide more experimental details.

5. In terms of writing, it should be carefully improved. For example, should the b1,b2,b3 in Figure 1 described in lines 118 to 123 of the article be c1,c2,c3? The format of lines 330-334 is very non-standard.

**Suitability:**

3

---

### Official Review · Reviewer_TDu3 · 2024-05-29

**Rating:** 4
**Confidence:** 4

**Summary:**

Two-stage pipeline:

The first stage is the restoration stage. It has a dual-branch encoder and a single-branch event decoder. The degraded events are fed into the event encoder while the blurry image is fed into the image encoder. Then, the features of the image encoder guide the event restoration process.

The output is the restored events.

The second Stage is the deblurring stage. It also has a dual-branch encoder and a single-branch image decoder. The restored events are used to guide the image deblurring task.

The output is the deblurred image.

**Strengths:**

(i) A great study is presented in the related work section where whatever work has been done till now is reviewed related to the event simulator and motion deblurring.

(ii) In Spite of the fact that the paper is using an event simulator that is generating the paired sets of {undegraded event, degraded event}, it has presented great detail on the three degradations that are introduced due to the DVS.

**These degradations are simulated by the event simulator.

(iii) The Two-Stage pipeline section is greatly explained and also a clear explanation of loss functions is there.

(iv) Results are well presented. Sufficient qualitative and quantitative analysis is provided.

**Limitations:**

**The title of the paper involves the term “Real-World Degraded Events”, but the entire model training is happening on simulated data.
The CVPR 2021 paper “Motion Deblurring with Real Events” addressed the issue and proposed a semi-supervised model trained on real data.
(**Reference not provided in the paper)
Link: https://arxiv.org/pdf/2109.13695

The paper we are reviewing evaluated all the other models by training them on the simulated data. However since the main motive of this paper is addressing the real-world degradations, it should present a study showing how the synthetically simulated degraded events are different from the real-world degraded events and the effect of this on the trained model. Since no proper explanation is presented, the practical application is questionable.

**The paper presented an understanding of the different degradations introduced by DVS but no analysis of how these are added to generate the paired set.

**Further, no mathematical analysis is presented in the entire paper.

**The explanation I found the paper is lacking is about the generation of complex degradations for the natural scene in the DAVISMCR dataset.

**Suitability:**

3

---

### Meta-Review · Area_Chair_87Gg · 2024-07-05

**Recommendation:** Accept (Poster)
**Confidence:** 5

**Metareview:**

This paper received mixed scores in the first round of review: 1 Weak Reject, 2 Borderline Accept, 1 Weak Accept. After rebuttal, the reviewer with Weak Reject slightly raised the score to Borderline Reject, and the other two reviewers remain unchanged. Positive reviews praised the problem setup and the dataset prepared. Yet, negative reviews have concerns on the gap between real data and synthetic data, and experiment comparision. Given the tight condition, AC carefully examined all materials at hand, and believes that this paper has merits. Yet, considering the remaining issues, AC discussed this paper with SAC in depth, finally decided to accept it. The authors are required to enhance the paper as they said in the rebuttal.